# Effect of the Thickness of TiO_2_ Films on the Structure and Corrosion Behavior of Mg-Based Alloys

**DOI:** 10.3390/ma13051065

**Published:** 2020-02-28

**Authors:** Aneta Kania, Piotr Nolbrzak, Adrian Radoń, Aleksandra Niemiec-Cyganek, Rafał Babilas

**Affiliations:** 1Department of Engineering Materials and Biomaterials, Faculty of Mechanical Engineering, Silesian University of Technology, Konarskiego 18a, 44-100 Gliwice, Poland; rafal.babilas@polsl.pl; 2HTL-Strefa S.A., Adamowek 7, 95-035 Ozorkow, Poland; nolbrzak.piotr@gmail.com; 3Łukasiewicz Research Network, Institute of Non-Ferrous Metals, Sowinskiego 5, 44-100 Gliwice, Poland; adrianr@imn.gliwice.pl; 4Professor Zbigniew Religa Foundation of Cardiac Surgery Development, Bioengineering Laboratory, Wolności 345a, 41-800 Zabrze, Poland; acyganek@frk.pl

**Keywords:** TiO_2_ films, Mg-based alloys, structure study, electrochemical study, hydrogen evolution, cytotoxicity assays

## Abstract

This article discusses the influence of the thickness of TiO_2_ films deposited onto MgCa2Zn1 and MgCa2Zn1Gd3 alloys on their structure, corrosion behavior, and cytotoxicity. TiO_2_ layers (about 200 and 400 nm thick) were applied using magnetron sputtering, which provides strong substrate adhesion. Such titanium dioxide films have many attractive properties, such as high corrosion resistance and biocompatibility. These oxide coatings stimulate osteoblast adhesion and proliferation compared to alloys without the protective films. Microscopic observations show that the TiO_2_ surface morphology is homogeneous, the grains have a spherical shape (with dimensions from 18 to 160 nm). Based on XRD analysis, it can be stated that all the studied TiO_2_ layers have an anatase structure. The results of electrochemical and immersion studies, performed in Ringer’s solution at 37 °C, show that the corrosion resistance of the studied TiO_2_ does not always increase proportionally with the thickness of the films. This is a result of grain refinement and differences in the density of the titanium dioxide films applied using the physical vapor deposition (PVD) technique. The results of 24 h immersion tests indicate that the lowest volume of evolved H_2_ (5.92 mL/cm^2^) was with the 400 nm thick film deposited onto the MgCa2Zn1Gd3 alloy. This result is in agreement with the good biocompatibility of this TiO_2_ film, confirmed by cytotoxicity tests.

## 1. Introduction

Biodegradable magnesium alloys have many advantages in comparison with other biomaterials used for bone implants. The density, modulus of elasticity (Young’s modulus), and tensile strength of magnesium are similar to human bones. Implants based on titanium, cobalt alloys, or stainless steel are used in orthopedic applications at present. They have many advantageous properties and are characterized by good corrosion resistance. However, these materials have to remain in the human body for many years. Magnesium and its alloys are characterized by in vivo biodegradability, and this advantage makes Mg alloys differ from the implants used today [1,2,3,4,5]. Therefore, Mg alloys can be used as temporary implants. An important property is the biocompatibility of magnesium, which means the acceptance of Mg by the surrounding tissues. In the human body, magnesium ions interact with polyphosphate compounds (e.g., adenosine triphosphate (ATP), DNA, and RNA). Moreover, Mg influences cell membrane permeability and lipid metabolism. For adults, the recommended daily allowance (RDA) of magnesium is 300–400 mg per day [2,3,4]. Additionally, the assimilability of magnesium is 40% of the consumed dose [3,4]. 

Nevertheless, the corrosion process of Mg is too fast, limiting the use of magnesium alloys in orthopedics. Moreover, the problem is that the high volume of evolved hydrogen during the corrosion process is related to the formation of hydroxide ions, resulting in an increase in the physiological pH next to the implant environment [5]. The H_2_ bubbles from corroding Mg alloys can lead to inflammation within the implant and tissue necrosis. Therefore, appropriate alloying elements are used [6,7]. It is important to avoid toxic elements during the selection of a biomaterial’s chemical composition. An increase in degradation rate causes an increase in the amount of contaminants that penetrate the tissues over time [8,9,10]. The reactions of the body’s defense system, the aggressiveness of the body fluids, the type of work performed by the biomaterial, the load on the material, and its physicochemical and electrical properties influence the implants’ degradation [11,12,13].

Mg-Zn-Ca alloys consist of elements that are widely found in the body and participate in many life processes [14,15,16]. Therefore, the corrosion process of implants made of these alloys and the elements’ dissolution do not disturb the body’s homeostasis [17]. The addition of an acceptable content of gadolinium into Mg alloys does not adversely influence the physiological environment [18,19,20]. Kubasek et al. [18] examined the differences in corrosion resistance and structure in the tissue environment between Mg-1Zn, Mg-3Zn, Mg-1Zn-3Gd, and Mg-3Zn-3Gd alloys. They stated that the addition of 1 and 3 wt % Zn improves the corrosion resistance of the binary alloys, while the addition of Gd into Mg-1Zn alloy decreases the corrosion rate. 

The corrosion behavior of magnesium and its alloys is a key issue in the search for different protective coatings [21,22,23,24,25,26,27,28]. Oxide films, such as MgO, ZnO, and TiO_2_, are deposited onto Mg alloys to prevent a high H_2_ evolution. Among these coatings, TiO_2_ coatings have many suitable properties for medical applications (e.g., high biocompatibility, biotolerance, corrosion resistance, etc.). Liberini et al. [27] studied the electrochemical behavior of four TiO_2_ nanoparticle coatings on AA2024 aluminum alloy. The authors obtained homogeneous, nanometric titania coatings by aerosol flame synthesis. The corrosion tests using electrochemical impedance spectroscopy were performed after immersion in de-aerated 0.5 M Na_2_SO_4_ solution at room temperature for 90 min. Results show that TiO_2_ coatings significantly improve the electrochemical properties of aluminum surfaces. Liu et al. [28] examined nano-TiO_2_ coatings produced by vacuum dip-coating method on pure and anodized aluminum surfaces. The authors studied the corrosion behavior of the films at 25 °C in sterile seawater. They stated that TiO_2_ coatings show excellent anticorrosion properties. Moreover, titanium dioxide is a bioactive and non-resorbable material. When in contact with body fluids, a hydroxyl apatite (HA) layer is formed on the surface of TiO_2_ [21]. Titanium dioxide naturally occurs in three main phases: anatase, rutile, and brookite. Anatase and rutile have good blood compatibility. Moreover, anatase is less cytotoxic and has antibacterial properties [29].

In vitro studies carried out by Amaravathy et al. [23] showed that an HA/TiO_2_-coated alloy was characterized by a higher osteoinduction compared to an HA-coated alloy. Moreover, the results of electrochemical and immersion tests confirmed a decrease in the degradation rate of the alloy with the HA/TiO_2_ coating. Cell culture tests also showed higher cell proliferation on the alloy with the composite coating [23].

The present paper describes the effect of the thickness of TiO_2_ films applied on MgCa2Zn1 and MgCa2Zn1Gd3 alloys using magnetron sputtering on their structure, corrosion behavior, and cytotoxicity.

## 2. Materials and Methods

MgCa2Zn1 and MgCa2Zn1Gd3 alloys were obtained by induction casting at 750 °C with Ar as the protective gas. The molten alloys from chamotte-graphite crucibles were cast into sand molds.

Samples of the studied alloys in the form of cylinders with diameters of 13 mm and heights of 4 mm were the substrate materials for the magnetron sputtering. The process was carried out at a temperature of 100 °C, with an Ar atmosphere using a Kurt J. Lesker PVD 75 device (Kurt J. Lesker Company, Jefferson Hills, USA). A Ti target (99.9% pure) was used for the magnetron sputtering process. Oxygen (99.99% pure) was supplied to the chamber. The process was performed over either 90 or 180 min. 

The thickness of the TiO2 films was measured using a Filmetrics F20 reflectometer (KLA Company, San Diego, USA). This device takes the reflection of light from a thin layer, and analyzes this light over a certain range of wavelengths. The reflectometer measured thickness in the range of 15 nm–70 µm.

### 2.1. Structure Study and Phase Analysis

The observations of surface morphology were conducted with a Zeiss SUPRA 35 scanning electron microscope (SEM, Thornwood, New York, USA) (EHT = 3.0 kV; SE mode, in-lens detector), which was equipped with an energy-dispersive (EDS) detector. 

Surface topography observations and roughness measurements were carried out using an AFM XE-100 atomic force microscope from Park Systems (Suwon, South Korea). The experiment was run in non-contact mode. The observation area was 25 µm^2^. XEI software (1.8 version) was used for the roughness parameters (e.g., roughness average (Ra) and root mean square (RMS)) calculations.

The phase analysis of the TiO2 films was carried out with a PANalytical X’Pert PRO X-ray diffractometer (PANalytical, Almelo, Netherlands), using Co Kα radiation. The analysis was conducted with the step registration method over a 2θ angular range from 25° to 80°. X-ray qualitative analysis was carried out with HighScore Plus software (3.0e version) using a dedicated PAN-ICSD (Inorganic Crystal Structure Database) phase identification card database.

### 2.2. Electrochemical and Immersion Tests

Electrochemical corrosion tests were performed using an Autolab PGSTAT302N Multi BA potentiostat (Metrohm AG, Herisau, Switzerland). A saturated calomel electrode was the reference electrode and a platinum rod was the counter electrode. The experiment was carried out in Ringer’s chloride-rich solution at a temperature of 37 °C. The scan rate of the corrosion potential was 1 mV·s^−1^. Before the measurements, the samples were immersed in Ringer’s solution for 5 min for stabilization. The corrosion parameters (e.g., corrosion potential—E_corr_, corrosion current density—i_corr_, and corrosion polarization resistance—R_p_) were determined using Tafel’s analysis.

Moreover, immersion tests for the estimation of gas corrosion product (volume of H_2_ evolution) for the TiO_2_ films were performed. The studies were conducted in the Ringer’s solution at 37 °C for 24 h.

Cylindrical samples, with a testing area of 1.1 cm^2^ for electrochemical studies, and of 1.3 cm^2^ for immersion tests, were prepared. The volume of H_2_ evolution was measured and calculated, taking into account the frontal area of the samples.

### 2.3. Analysis of Corrosion Products

After the immersion tests, the corroded surfaces of the TiO_2_ films were observed using SEM. To remove the corrosion products, the samples were rinsed with distilled water and immersed in CrO_3_ solution before the observations.

Fourier transform infrared (FTIR) spectroscopy was used to analyze the corrosion products. FTIR spectra were recorded for the MgCa2Zn1 and MgCa2Zn1Gd3 alloys with the TiO_2_ films at room temperature using a Nicolet 6700/8700 FTIR spectrometer (Thermo Fisher Scientific, Waltham, USA). For this purpose, the corrosion products were collected from the surface of immersed samples and mixed together with dry KBr. Measurements of samples were conducted in transmission mode in a mid infrared range of 4000–400 cm^−1^.

### 2.4. Cytotoxicity Assays

Preliminary biological in vitro investigations were carried out using mouse fibroblast cells. Samples of the MgCa2Zn1 and MgCa2Zn1Gd3 alloys in their as-cast state and with TiO_2_ films were used for the experiments. The samples were sterilized in an antibiotic bath. Cell culture was carried out in Medium 199 with 10% fetal bovine serum (FBS) over 24 h. The tests were performed at 37 °C with a CO_2_ (5%) atmosphere and a constant 95% humidity. The samples were placed into a dish culture, where the confluence of the cells was up to 90%. The lack of vital functions (necrosis) of the cells was estimated after 24 h. FDA (fluorescein diacetate) was used as the cell viability stain, and propidium iodide (PI) was used to determine the number of dead cells. The relative number of dead cells was measured using a Zeiss AXIO OBSERVER (Zeiss, New York, USA) inverted fluorescence microscope equipped with AxioVision 4.6 software.

## 3. Results and Discussion

MgCa2Zn1 and MgCa2Zn1Gd3 alloys were used as substrate materials for the magnetron sputtering process. The titanium dioxide films were applied onto the two Mg-based alloys with 90 and 180 min deposition times.

The thicknesses of the TiO_2_ films measured with the reflectometer were 200 nm (for the deposition time of 90 min) and 400 nm (for the 180 min deposition time), respectively. The results of the measurements showed that the thickness of the films increased proportionally to the increase in the deposition time. The 200 nm thick TiO_2_ film was selected based on previous studies presented by Kalisz et al. [30]. The authors applied TiO_2_ film (200 nm thick) onto biomedical Ti6Al4V alloy using conventional magnetron sputtering process. The thin film had an anatase structure. They stated that the TiO_2_ coating showed the best corrosion properties among other tested coatings in 0.5M NaCl solution. Based on these results, the 200 nm thick film was applied onto magnesium alloys to check the corrosion behavior.

The TiO_2_ films applied onto the MgCa2Zn1 and MgCa2Zn1Gd3 alloys were similar in surface morphology, the grains of the films being formed with a spherical-like morphology (Figure 1) [21]. In the case of oxide films deposited onto the MgCa2Zn1 alloy, the grains were slightly different in sizes. The surface of the 200 nm thick TiO_2_ was characterized by grains with sizes in the range of 18–120 nm (Figure 1a). The second film, with a thickness of about 400 nm, was characterized by a larger number of grain aggregations with defined grain boundaries. The grain sizes were from 18 to 160 nm (Figure 1b). The results of porosity measurements showed that the porosity of the thinner TiO_2_ film was equal to 1.85% (±0.7 SD), and of the 400 nm thick film was 3.56% (±1.1 SD). It can be stated that for the thickness of 200 nm and the MgCa2Zn1 alloy as a substrate material, the TiO_2_ film was denser and more compact compared to the thicker TiO_2_ film. Similar surface morphologies to the 400 nm thick TiO_2_ film were noted with the titanium dioxide films deposited onto the MgCa2Zn1Gd3 alloy (Figure 1c,d). It could be observed that the grains of the 200 and 400 nm thick TiO_2_ films also had similar sizes to the grains of the 400 nm thick film deposited on the MgCa2Zn1 alloy. The porosity of the thinner film was equal to 3.43 (±1.4 SD), and for the 400 nm thick film was 2.91% (±1.2 SD).

The results of X-ray phase analysis confirmed that particular TiO_2_ phases were detected in the studied films (Figure 2 and Figure 3). In the diffraction patterns, there were characteristic peaks at 29.47, 44.17, 56.43, 63.49, and 64.94 degrees of the 2θ angle for both the TiO_2_ deposited onto MgCa2Zn1 and that on the MgCa2Zn1Gd3. Based on the phase analysis, it can be stated that the TiO_2_ has an anatase structure (space group I41/amd; space group No. 141; lattice parameters: a = 3.784 Å, b = 3.784 Å, c = 9.515 Å). Anatase has a tetragonal crystal structure [24]. This structure is more biologically active compared to other structures, such as rutile or brookite. Moreover, the XRD patterns showed strong diffraction peaks from Mg on the alloy’s surface.

The analysis of the topography of the TiO_2_ films was performed in an area of 25 µm^2^. The films with a thickness of about 200 and 400 nm were similar in their topography for both the alloys. It could be observed that the studied TiO_2_ films had a granular structure (Figure 4). The aggregations of particles for the same TiO_2_ thickness for the both Mg alloys had a similar shape. Based on the analysis results for all the samples, it can be stated that the surface irregularities of the TiO_2_ films did not exceed 25 nm.

During the measurements, roughness parameters (e.g., roughness average, R_a_, and root mean square, RMS) were determined (Table 1). The results of the roughness measurements showed that the values of R_a_ and RMS for the 200 and 400 nm thick TiO_2_ applied onto the MgCa2Zn1 alloy were similar. The values of the roughness parameters decreased with the increase in the films’ thickness. The irregularities occurring on the surface of the thinner films were filled out during the deposition process of the 400 nm thick films, hence the differences in the roughness values. Larger differences in the R_a_ and RMS values occurred for the films deposited on the MgCa2Zn1Gd3 alloy.

Electrochemical investigations of the corrosion resistance of the alloys, with and without TiO_2_ films, were carried out. The measurements were performed in Ringer’s solution, which is enriched in chlorides, at a temperature of 37 °C. Such potentiodynamic curves are known to give some information on the corrosion mechanism of TiO_2_ films [5]. The titanium dioxide films deposited on the MgCa2Zn1 and MgCa2Zn1Gd3 alloys were characterized by higher values of corrosion potentials (E_corr_), and lower values of corrosion current densities (i_corr_) compared to the substrate materials (Figure 5 and Figure 6). This may suggest an improvement in the corrosion resistance of the studied alloys with TiO_2_ deposited by the PVD technique [5,17,31]. Li et al. [31] studied the corrosion behavior for titania coatings on Mg-Ca alloy immersed in SBF (simulated body fluid) solution for 48, 168, and 336 h. They stated that the corrosion resistance was improved for the alloy with TiO_2_.

The results of the electrochemical studies showed that the MgCa2Zn1Gd3 alloy with the 400 nm thick TiO_2_ film had a better corrosion resistance compared to the 200 nm thick film (Figure 5). This result was in agreement with the results of roughness measurements, where the RMS and R_a_ values of this film were lower compared to the thinner film. The corrosion parameters for the both the studied alloys and the TiO_2_ films applied are listed in Table 2. The corrosion potential of the thicker film shifted to more positive values (E_corr_ for the 200 nm thick TiO_2_ film was −1.50 V, and E_corr_ for the 400 nm thick TiO_2_ film was −1.43 V). The values of corrosion current densities and polarization resistances (R_p_) for the studied films also confirmed an improvement in the corrosion resistance of the MgCa2Zn1Gd3 alloy (i_corr_ for the TiO_2_/200 nm was 130 μA·cm^−2^, and R_p_ was 440 Ω·cm^2^; i_corr_ for the TiO_2_/400 nm was 98 μA·cm^−2^, and R_p_ was 510 Ω·cm^2^).

In the case of the oxide films deposited onto the MgCa2Zn1 alloy, differences in the corrosion potential values compared to the films applied on MgCa2Zn1Gd3 were observed (Figure 6). The E_corr_ for the TiO_2_ film with a deposition time of 90 min was −1.40 V, and for the thicker one was −1.43 V. The values of R_p_ were 350.4 and 298.6 Ω·cm^2^, for the 200 and 400 nm thick TiO_2_ films, respectively. The corrosion current density for the 200 nm thick TiO_2_ film was 78 μA·cm^−2^, and for the thicker film it was 96 μA·cm^−2^. The results of the electrochemical tests indicated that the increase in TiO_2_ thickness did not always decrease the degradation rate. The higher density of the thinner film affected the corrosion behavior by improving the corrosion resistance of the substrate material [17,31].

The results of the roughness studies for the both TiO_2_ films show that the RMS and R_a_ values of the 200 nm thick film were slightly higher. In some reports on surface treatments [32], the authors stated that there was no influence of the roughness on the degradation rate of the alloys with deposited coatings. Gawlik et al. [32] indicated that in some cases rougher films are characterized by less pitting corrosion and quite good cell adhesion. However, electrochemical results are the only available information about the initial state of the corrosion behavior [33].

After the potentiodynamic tests, the studied TiO_2_ films and uncoated alloys were immersed in Ringer’s solution at a temperature of 37 °C, over a time of 24 h. The results of such immersion tests indicate the real long-term corrosion behavior of materials for medical applications [7,33]. After 24 h of immersion, the 400 nm thick film deposited onto MgCa2Zn1 alloy was characterized by a higher H_2_ volume of 12.4 mL/cm^2^. The volume of hydrogen evolution for the thinner film was 8.8 mL/cm^2^ (Figure 7). The corrosion rates, v_corr_, calculated from the hydrogen evolution volume after 24 h of immersion were 3.08 and 4.34 mm·y^−1^ for the 200 and 400 nm thick TiO_2_, respectively. The v_corr_ for the MgCa2Zn1 alloy was 4.9 mm·y^−1^. The results of immersion tests confirmed the electrochemical results, where the thinner film was characterized by a slight improvement in the corrosion resistance. This probably resulted from the grain refinement of the 200 nm thick TiO_2_ film. Moreover, this film was denser and more complex compared to the thicker one.

A slightly lower volume of evolved hydrogen was noted for the films applied onto MgCa2Zn1Gd3 alloy (Figure 8). It can be observed that the volume of H_2_ evolved decreased almost proportionally with respect to the increase in the thickness of the TiO_2_ films.

The volume of evolved H_2_ was 11.85 and 5.92 mL/cm^2^ for the 200 and 400 nm thick films, respectively. The corrosion rate for the thinner film was 4.81 mm·y^−1^, and for the thicker one was 2.4 mm·y^−1^. Moreover, the 400 nm thick TiO_2_ film was characterized by the lowest volume of evolved hydrogen compared to other studied films. This result bodes well for future investigations. The vcorr for the MgCa2Zn1Gd3 alloy was 6.45 mm·y^−1^. Amaravathy et al. [23] examined the hydrogen evolution of HA/TiO_2_ on Mg alloy. They confirmed the decrease in corrosion rate for the composite coated alloy by the reduction in the evolved hydrogen produced by magnesium corrosion. The hydrogen evolution process was also analyzed by Bakhsheshi-Rad et al. [34]. The H2 evolution rates of Si/TiO_2_, and Si coated Mg-Ca alloys were 1.57 mL/cm^2^/day and 2.22 mL/cm^2^/day, respectively. The uncoated alloy was characterized by a higher hydrogen evolution rate of 5.04 mL/cm^2^ per day.

After the immersion tests, the samples of the studied alloys with TiO_2_ films with improved corrosion resistance were microscopically observed. The results of the microscopic observations of the 200 nm thick film deposited onto MgCa2Zn1 and of the 400 nm thick film applied on the MgCa2Zn1Gd3 alloy are presented in Figure 9.

Microcracks were observed in both the TiO_2_ films, due to dehydration during the drying of the samples (Figure 9a,b). Some needle-like and lamellar-shaped corrosion products were visible on the surfaces of the titanium dioxide films (Figure 9c,d). These were probably from magnesium hydroxides [35,36]. Mg(OH)_2_ is the main corrosion product of Mg alloys. It forms a loose film and cannot protect the Mg alloy surface from the degradation process [37]. In our study, it was observed that some corrosion products fell off from the surfaces of the films (Figure 9c,d). After the removal of the corrosion products, the grain boundaries were still visible (Figure 9e,f). Small pits were observed on the surfaces of both the oxide films, which may suggest the occurrence of pitting corrosion [33]. It can be stated that the corroded surface of the TiO_2_ film deposited onto MgCa2Zn1Gd3 alloy was characterized by lower corrosion damage compared to the second film (Figure 9e). This confirmed an improvement in the corrosion resistance of the film applied onto the Gd-containing alloy. Although this 400 nm TiO_2_ film is uniform, it is porous. The chloride ions in Ringer’s solution (8.6 g/dm^3^ NaCl, 0.3 g/dm^3^ KCl, 0.48 g/dm^3^ CaCl_2_ 6H_2_O) penetrate the TiO_2_ coating and reach the substrate through the micropores. Increase in immersion time causes that the TiO_2_ film begins to degrade [34]. Thus more solution interacts with the Mg alloy. When TiO_2_ coated alloy is immersed in Ringer’s solution, the MgO in the outer layer starts to react with chloride-containing solution, and converts to Mg(OH)_2_.

The corrosion products of the TiO_2_ films were characterized by FTIR spectroscopy. The analysis results, presented in Figure 10, confirmed the presence of both hydroxides and carbonates as the main corrosion products. For all samples, the broad band in the range of 2700 cm^−1^ to 3600 cm^−1^ and a weak peak at 1645 cm^−1^ were related to the O-H-O and O-H vibrations from water. The sharp peaks at 3700 cm^−1^ and at 3645 cm^−1^ were associated with the stretching mode of the O-H vibration in, especially, Mg(OH)_2_. Moreover, a broad peak in the range of 1291 cm^−1^ to 1581 cm^−1^ was due to the presence of carbonates, such as MgCO_3_. Additionally, the FTIR spectrum also exhibited symmetric stretching vibrations from Mg-O (444 cm^−1^) and Zn-O or Ti-O (544 cm^−1^) [38]. According to these results, the main corrosion reactions were associated with the formation of hydroxides and carbonates. It can be assumed that the hydroxides were formed first, and then the reaction between them and CO_2_ from the air resulted in the formation of carbonates. The presence of Mg(OH)_2_ confirmed that the needle-like and lamellar-shaped corrosion products were from magnesium hydroxides.

Measurements of the relative number of dead cells after 24 h of culture for the studied alloys and the alloys with TiO_2_ films were conducted (Figure 11). The investigations included only the measurements of the titanium dioxide films with the lowest volume of evolved H_2_ from the immersion tests. The hydrogen evolution volume has a strong impact on cell viability and proliferation. The investigation results showed that the MgCa2Zn1 alloy and this alloy with the 200 nm thick TiO_2_ film caused a toxic effect on the cell culture compared with the control sample. The number of dead cells was equal to 38% (±4.2 SD) and 32% (±3.6 SD) for the MgCa2Zn1 and the alloy with the 200 nm thick film, respectively. This indicates a cytotoxic effect in accordance with ISO 10993-5 standard [39]. These materials also caused very strong cell lysis (cell disintegration due to the destruction of cell membranes) (Figure 11a). Moreover, changes in cells morphology and proliferation were shown. The cell density was low, most of the cells were rounded, what indicates their degradation. On the other hand, the gadolinium-containing alloy was characterized by less cytotoxicity, the number of dead cells being equal to 28% (±5 SD). The cytotoxicity of the MgCa2Zn1Gd3 was found to be grade 2, which means a mild reactivity (Figure 11b). The achievement of a numerical grade greater than 2 indicates a cytotoxic effect [39]. This material was characterized with slightly lower cell density, the morphology of cells seeded on this substrate was correct. A lower cell necrosis was observed for the 400 nm thick TiO_2_ deposited onto MgCa2Zn1Gd3 alloy, the number of dead cells being equal to 18% (±3.7 SD) in comparison with the control sample (Figure 11c).

The titanium dioxide film was characterized by a slight reactivity, achieving a grade of 1 [39]. This result was in agreement with the immersion test results, where the volume of evolved H_2_ for the 400 nm thick TiO_2_ film was 5.92 mL/cm^2^. The density of cells cultured on this material was compared to control. The cells had proper, characteristic to fibroblasts, spindle-shaped morphology.

The results of the cytotoxicity tests for the MgCa2Zn1Gd3 alloy with 400 nm thick titanium dioxide film are promising for future research.

Cell culture studies were also performed by Amaravathy et al. [40]. They stated that a TiO_2_ coated AZ31 alloy showed higher cell attachment and proliferation compared to the uncoated alloy.

## 4. Conclusions

The analysis of the investigation’s results showed that the surface morphology of the studied TiO_2_ films was homogeneous, with grains of a spherical shape and dimensions from 18 to 160 nm. Slight differences in the morphology were observed for the 200 nm thick TiO_2_ film applied onto MgCa2Zn1 alloy. This film was denser and more complex compared to the other studied films. Based on the XRD analysis, it can be stated that all the titanium dioxide films had an anatase structure, which is known to have antibacterial properties. The results of the Tafel’s analysis showed that the oxide films slightly improved the corrosion resistance of the studied MgCa2Zn1 and MgCa2Zn1Gd3 alloys. The corrosion resistance of the TiO_2_ films applied onto MgCa2Zn1Gd3 was slightly improved by increasing the thickness of the films. In the case of the oxide films deposited on the MgCa2Zn1 alloy, the results of electrochemical and immersion tests differed. The 200 nm thick titanium dioxide film was characterized by a higher corrosion resistance compared to the thicker film. This was because of the grain refinement of the thinner TiO_2_ film. The films deposited onto the MgCa2Zn1Gd3 alloy showed a lower volume of hydrogen evolution in comparison to the TiO_2_ applied on the MgCa2Zn1 alloy. After 24 h of immersion, the volume of evolved H_2_ for the 400 nm thick TiO_2_ film deposited onto MgCa2Zn1Gd3 was equal to 5.92 mL·cm^−2^, and for the thinner film was 11.85 mL·cm^−2^. The hydrogen evolution volumes for the oxide films applied onto the MgCa2Zn1 alloy were 12.4 and 8.8 mL·cm^−2^ for the 400 and 200 nm thick TiO_2_, respectively. The results of the cytotoxicity tests confirmed that the titanium dioxide film (400 nm thick) deposited on the magnesium alloy with gadolinium addition fostered cell proliferation. The number of dead cells calculated using cytotoxicity tests was equal to 18%. This result may suggest a good biocompatibility of this TiO_2_ film.

## Figures and Tables

**Figure 1 materials-13-01065-f001:**
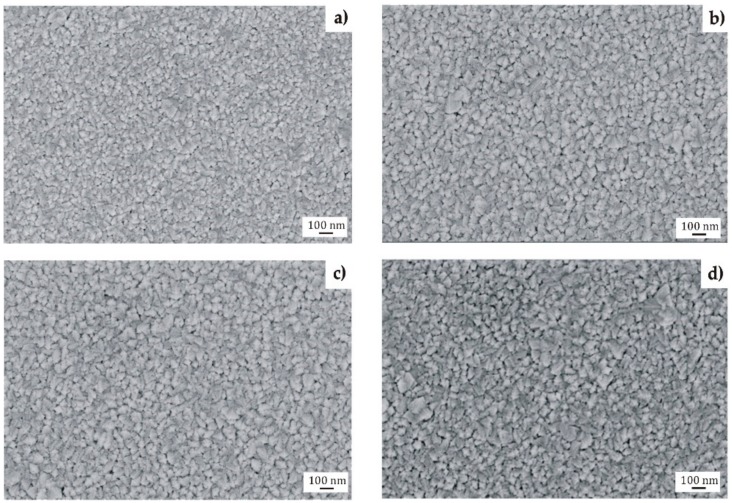
SEM images of the TiO_2_ surface morphology deposited on MgCa2Zn1: (**a**) 200 nm thick; (**b**) 400 nm thick, and MgCa2Zn1Gd3: (**c**) 200 nm thick; (**d**) 400 nm thick.

**Figure 2 materials-13-01065-f002:**
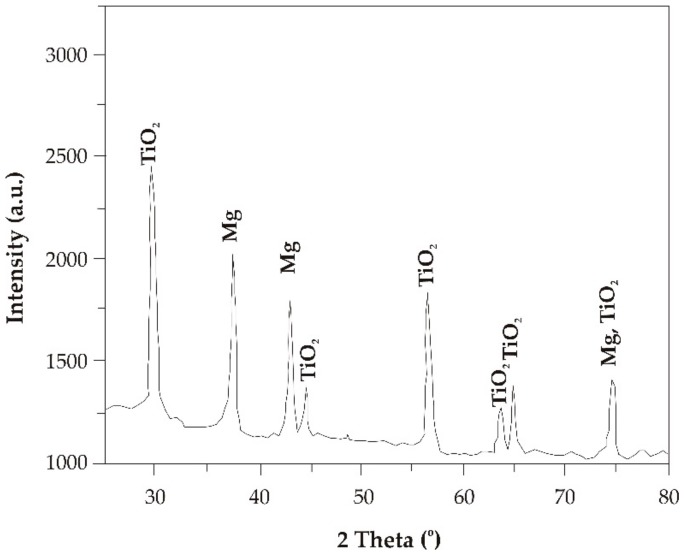
X-ray diffraction pattern of the TiO_2_ film (400 nm thick) applied onto MgCa2Zn1Gd3 alloy.

**Figure 3 materials-13-01065-f003:**
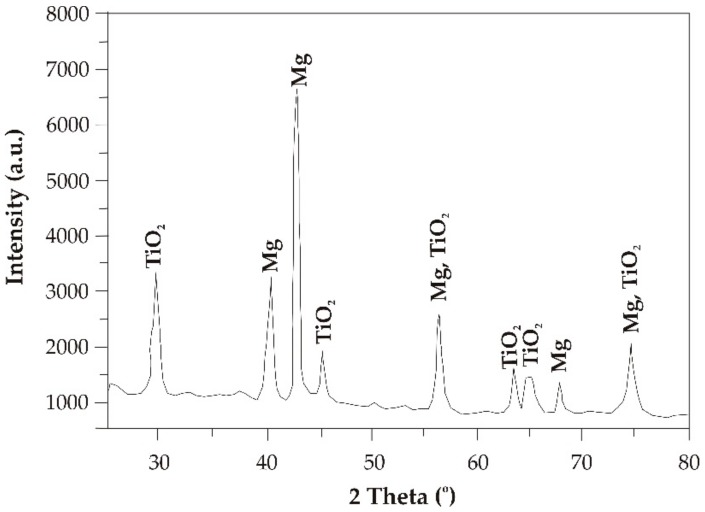
X-ray diffraction pattern of the TiO_2_ film (400 nm thick) applied onto MgCa2Zn1 alloy.

**Figure 4 materials-13-01065-f004:**
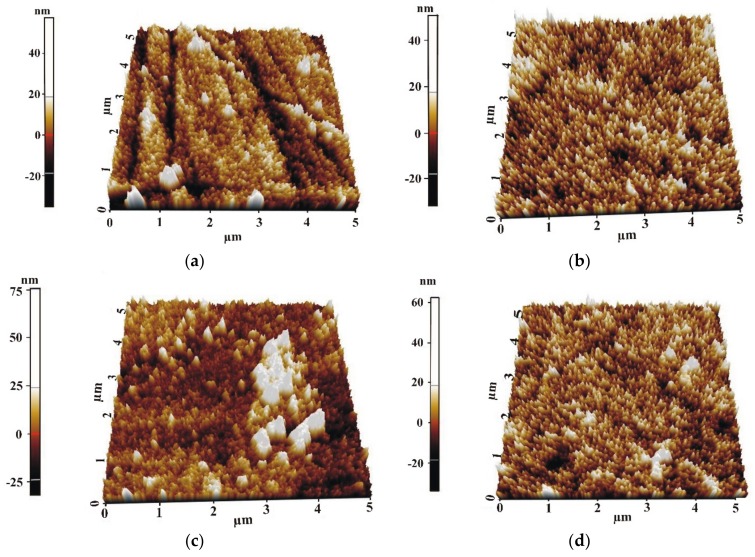
The AFM images of the 3D rendering of the TiO_2_ surface topography deposited onto MgCa2Zn1: (**a**) 200 nm thick; (**b**) 400 nm thick, and MgCa2Zn1Gd3: (**c**) 200 nm thick; (**d**) 400 nm thick.

**Figure 5 materials-13-01065-f005:**
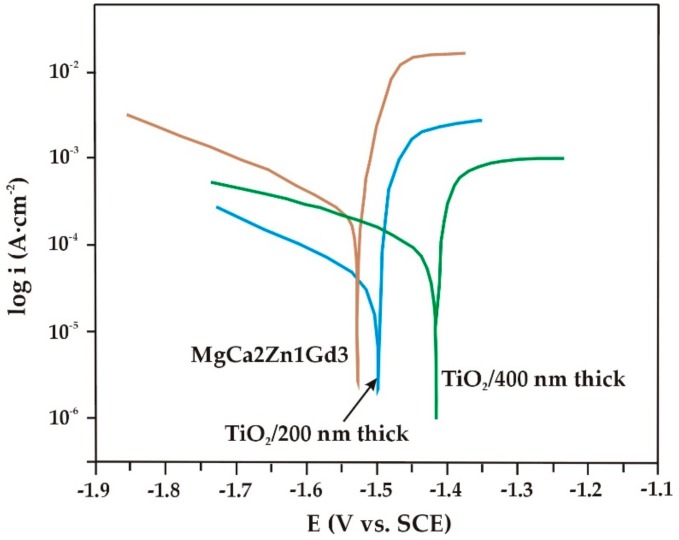
Polarization curves for the TiO_2_ films and MgCa2Zn1Gd3 alloy in Ringer’s solution at 37 °C.

**Figure 6 materials-13-01065-f006:**
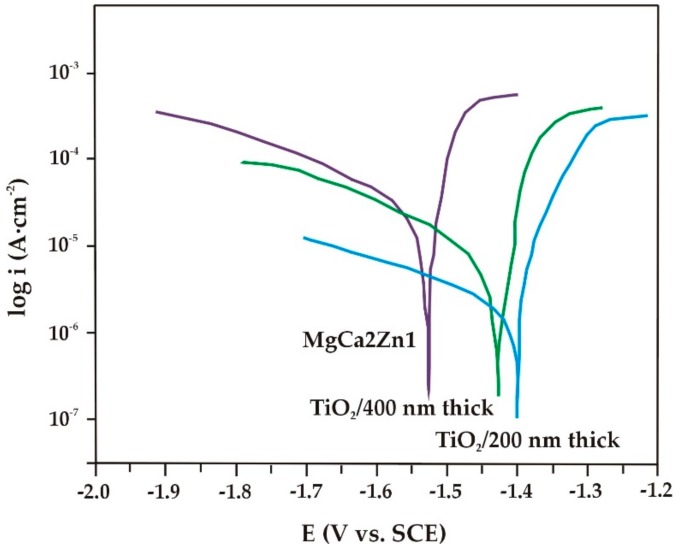
Polarization curves for the TiO_2_ films and MgCa2Zn1 alloy in Ringer’s solution at 37 °C.

**Figure 7 materials-13-01065-f007:**
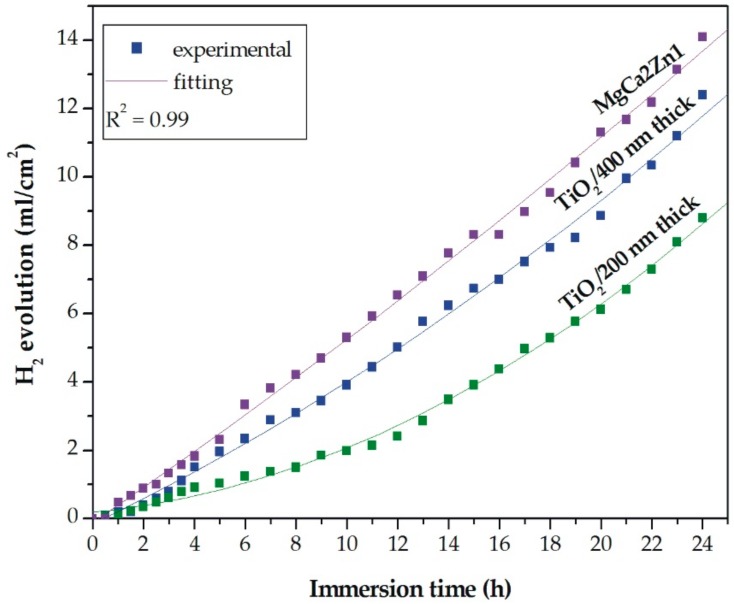
Volume of hydrogen evolution as a function of immersion time in Ringer’s solution at 37 °C during 24 h for the studied TiO_2_ films applied onto MgCa2Zn1 alloy and uncoated alloy.

**Figure 8 materials-13-01065-f008:**
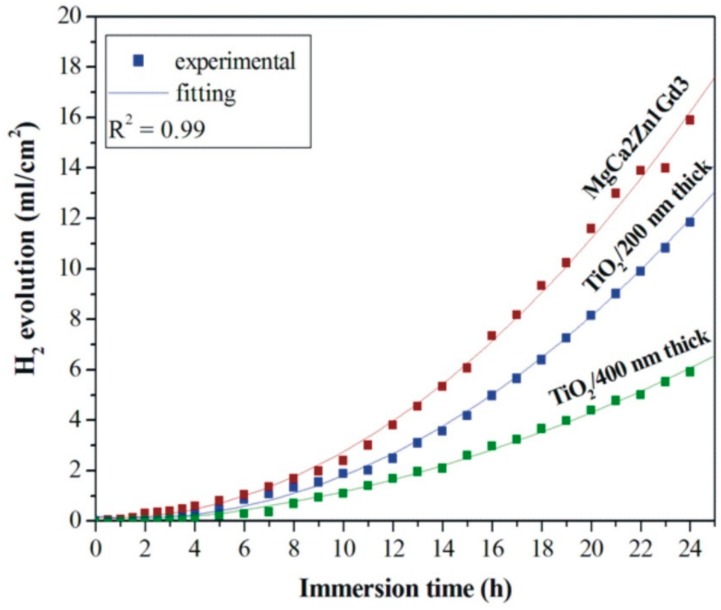
Volume of hydrogen evolution as a function of immersion time in Ringer’s solution at 37 °C during 24 h for the studied TiO_2_ films applied onto MgCa2Zn1Gd3 alloy and uncoated alloy.

**Figure 9 materials-13-01065-f009:**
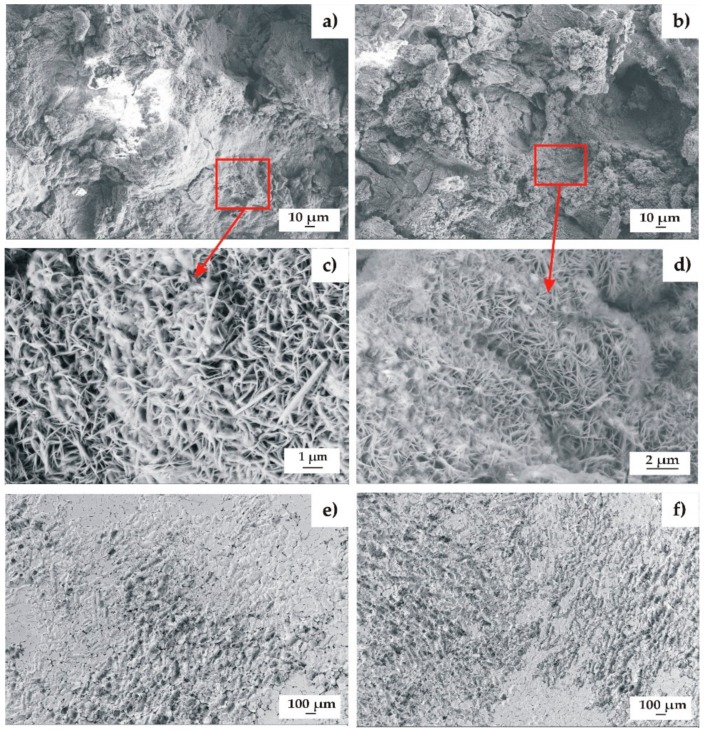
SEM images of samples’ surfaces with corrosion products (**a**–**d**) and without corrosion products (**e**,**f**) of the TiO_2_ films applied onto: (**a**,**c**,**e**) MgCa2Zn1Gd3 alloy; (**b**,**d**,**f**) MgCa2Zn1 alloy after 24 h of immersion in Ringer’s solution at 37 °C.

**Figure 10 materials-13-01065-f010:**
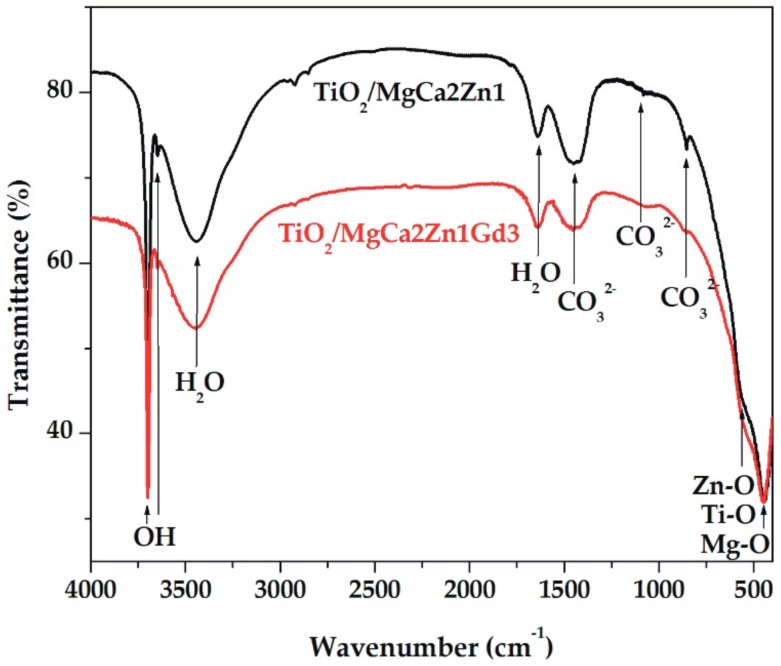
FTIR spectra of corrosion products collected from the surface of MgCa2Zn1 and MgCa2Zn1Gd3 alloys with deposited TiO_2_ films after 24 h of immersion in Ringer’s solution at 37 °C.

**Figure 11 materials-13-01065-f011:**

The effect of mouse fibroblast cultures exposed to materials for implant application: (**a**) 200 nm thick TiO_2_ film applied onto MgCa2Zn1 alloy; (**b**) MgCa2Zn1Gd3 alloy; (**c**) 400 nm thick TiO_2_ film applied onto MgCa2Zn1Gd3 alloy; dead cells are red; visible destruction of individual cells (cell lysis)–(**a**).

**Table 1 materials-13-01065-t001:** Surface roughness parameters of the TiO_2_ films applied, and MgCa2Zn1 and MgCa2Zn1Gd3 alloys.

Sample	Roughness Average, R_a_, nm	Root Mean Square, RMS, nm
MgCa2Zn1	4.60	5.91
MgCa2Zn1Gd3	4.86	5.43
TiO_2_ (200 nm thick)/MgCa2Zn1	7.28	9.60
TiO_2_ (400 nm thick)/MgCa2Zn1	7.18	9.45
TiO_2_ (200 nm thick)/MgCa2Zn1Gd3	8.90	12.30
TiO_2_ (400 nm thick)/MgCa2Zn1Gd3	7.38	9.48

**Table 2 materials-13-01065-t002:** Electrochemical parameters of MgCa2Zn1 and MgCa2Zn1Gd3 alloys, and the TiO_2_ films applied.

Sample	Corrosion Potential, E_corr_, V	Polarization Resistance, R_p_, Ω·cm^2^	Corrosion Current Density, i_corr_, μA·cm^-2^
MgCa2Zn1Gd3	−1.53 ± 0.03	384 ± 4	150 ± 3
TiO_2_ (200 nm thick)/MgCa2Zn1Gd3	−1.50 ± 0.03	440 ± 3	130 ± 3
TiO_2_ (400 nm thick)/MgCa2Zn1Gd3	−1.43 ± 0.03	510 ± 4	98 ± 2
MgCa2Zn1	−1.52 ± 0.03	286 ± 3	114 ± 5
TiO_2_ (200 nm thick)/MgCa2Zn1	−1.40 ± 0.03	350.4 ± 9	78 ± 2
TiO_2_ (400 nm thick)/MgCa2Zn1	−1.43 ± 0.03	298.6 ± 6	96 ± 3

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
