# Peer review of "Effect of the Thickness of TiO2 Films on the Structure and Corrosion Behavior of Mg-Based Alloys"

_materials, 2020, doi:10.3390/ma13051065_

Round 1
Reviewer 1 Report
The paper by Kania et al. presents method and characterization on the deposition of TiO2 films onto Mg-based alloy surfaces. The method is based on the well-known magnetron sputtering technique. The authors produce thin films having different thickness and surface morphology and characterize their properties in terms of improvement of corrosion behavior of the metal surfaces and biocompatibility of the films.
The paper is interesting and clearly presents the potential of the proposed TiO2 thin coatings. The morphology of the films, the corrosion properties of the coatings and the cytotoxicity of TiO2 films have been properly characterized by the authors. There are some revisions I’d like to recommend before considering the manuscript suitable for publication.
Particularly:
- The paper needs a text editing. There are some typos, and the English language needs to be improved.
- Concerning the references: the authors cite 37 papers in the whole document. In both the introduction section and the discussion section, literature regarding the anticorrosive properties of TiO2 thin has not been adequately reviewed by the authors. The authors should improve references by citing, e.g.: Liberini et al., Thin Solid Films 609 (2016) 53–61; T. Liu, et al. Surf. Coat. Technol. 205 (2010) 2335–2339. Those papers report the improvement of the electrochemical properties of aluminum surfaces achieved by applying thin coatings of TiO2 nanoparticles with different methods.
- The analysis performed by the authors on surface morphology by SEM and AFM is interesting and properly discussed. However, what is a bit puzzling is that they assume the 200 nm films to be denser and more compact than the 400 nm film by simply looking at the SEM images. Since the authors correlate the density of the films to some of their properties, a measurement of the porosity of the films should be provided.
Author Response
The authors want to thank you for the comments and suggestions to the manuscript. We tried to address all your remarks. Below, we are sending answers to the comments. We also revised the manuscript in accordance with the remarks.
Point 1: The paper needs a text editing. There are some typos, and the English language needs to be improved.
Response 1: The authors want to thank you for the remark. The text of the manuscript has been proofread by a native speaker of English who holds a PhD in Organic Chemistry. The authors attached the certificate of proofreading. Moreover, the text was checked again to eliminate some typing and grammar errors.
Point 2: The authors cite 37 papers in the whole document. In both the introduction section and the discussion section, literature regarding the anticorrosive properties of TiO2 thin has not been adequately reviewed by the authors. The authors should improve references by citing, e.g.: Liberini et al., Thin Solid Films 609 (2016) 53–61; T. Liu et al. Surf. Coat. Technol. 205 (2010) 2335–2339. Those papers report the improvement of the electrochemical properties of aluminum surfaces achieved by applying thin coatings of TiO2 nanoparticles with different methods.
Response 2: The authors agree with the Reviewer’s opinion. In the manuscript there are not so many references on anticorrosive properties of the TiO2 thin films. The authors improve the manuscript by citing Liberini et al., Thin Solid Films 609 (2016) 53-61 and Liu, et al. Surf. Coat. Technol. 205 (2010) 2335-2339.
“The corrosion behavior of magnesium and its alloys is a key issue in the search for different protective coatings [21–28]. Oxide films, such as MgO, ZnO, and TiO2, are deposited onto Mg alloys to prevent a high H2 evolution. Among these coatings, TiO2 coatings have many suitable properties for medical applications (e.g. high biocompatibility, biotolerance, and corrosion resistance, etc.). Liberini et al. [27] studied the electrochemical behavior of four TiO2 nanoparticle coatings on AA2024 aluminum alloy. The authors obtained homogeneous, nanometric titania coatings by aerosol flame synthesis. The corrosion tests using electrochemical impedance spectroscopy were performed after immersion in de-aerated 0.5M Na2SO4 solution at room temperature for 90 min. Results show the TiO2 coatings significantly improve the electrochemical properties of aluminum surfaces. Liu et al. [28] examined nano-TiO2 coatings produced by vacuum dip-coating method on pure and anodized aluminum surfaces. The authors studied the corrosion behavior of the films at 25 °C in sterile seawater. They stated that TiO2 coatings show excellent anticorrosion properties.”
Point 3: The analysis performed by the authors on surface morphology by SEM and AFM is interesting and properly discussed. However, what is a bit puzzling is that they assume the 200 nm films to be denser and more compact than the 400 nm film by simply looking at the SEM images. Since the authors correlate the density of the films to some of their properties, a measurement of the porosity of the films should be provided.
Response 3: The authors want to thank you for the remark. We agree that the measurements of the porosity of the films give some more information about the density of them. The measurements of the porosity of the TiO2 films were performed based on analysis of 10 SEM images with different magnification. The porosity of the thin films is presented in Table 1.
Table 1. Porosity of the TiO2 films.
|
TiO2 thickness |
Porosity, % |
|
TiO2 (200 nm thick)/MgCa2Zn1 TiO2 (400 nm thick)/MgCa2Zn1 TiO2 (200 nm thick)/MgCa2Zn1Gd3 TiO2 (400 nm thick)/MgCa2Zn1Gd3 |
1.85 ± 0.7 3.56 ± 1.1 3.43 ± 1.4 2.91 ± 1.2 |
[27] Liberini, M.; De Falco, G.; Scherillo, F.; Astarita, A.; Commodo, M.; Minutolo, P.; D’Anna, A.; Squilace, A. Nano-TiO2 coatings on aluminum surfaces by aerosol flame synthesis. Thin Sol. Films 2016, 609, 53–61, DOI: 10.1016/j.tsf.2016.04.025.
[28] Liu, T.; Zhang, F.; Xue, Ch.; Li, L.; Yin, Y. Structure stability and corrosion resistance of nano-TiO2 coatings on aluminum in seawater by a vacuum dip-coating method. Surf. Coat. Technol. 2010, 205, 2335–2339, DOI: 10.1016/j.surfcoat.2010.09.028.

Reviewer 2 Report
This manuscript shows a systematic investigation on the effect of the thickness of TiO2 layer with the biodegradable Mg alloys. Many necessary tests were conducted and well described.
It would be great if the authors provide a description why 200 and 400nm thick TiO2 were chosen for the tests. There should be stress mismatch if a thicker film is used. How this would be explained?
What would be the mechanism of the corrosion behavior of Mg with a thicker TiO2 film? Where the body fluids go if a high quality uniform thick TiO2 is covered?
It would be great if the authors show how the corrosion rates are changed with different thickness of films? With the hydrogen evolution reduction, there should be delayed Mg corrosion based upon the references cited in Page 10.
The cell culture study shows the improved cell viability with a thicker TiO2 film, which is well known outcomes from many previous studies. It would be great how the cell morphologies look like and how this study has its own originality.
Author Response
The authors want to thank Reviewer 2 for the comments and suggestions to the manuscript. We tried to address all your remarks. Below, we are sending answers to the comments. We also revised the manuscript in accordance with the remarks.
Point 1: It would be great if the authors provide a description why 200 and 400 nm thick TiO2 were chosen for the tests. There should be stress mismatch if a thicker film is used. How this would be explained?
Response 1: The authors want to thank you for the remark. The 200 nm thick TiO2 film was selected based on previous studies presented by Kalisz et al. [1]. The authors applied TiO2 film (200 nm thick) onto Ti6Al4V alloy using conventional magnetron sputtering process. The thin film had an anatase structure. They stated that the TiO2 coating showed the best corrosion properties among other tested coatings in 0.5M NaCl solution. Based on these results, we applied the 200 nm thick film onto magnesium alloys to check the corrosion behavior. Moreover, the authors wanted to determine the correlation between corrosion behavior and the thickness of the films. So, we selected the second thickness of the film equal to 400 nm. The presented studies were carried out to verify if the corrosion resistance increases proportionally to the thickness of the 200 and 400 nm thick TiO2 films.
The deposition process of the TiO2 films was preceded by deposition of Ti (99.9% pure) with an argon atmosphere for 2 minutes. The thickness of the applied Ti layer was from 20 to
50 nm. This thickness was selected experimentally. The titanium interlayer improves the adhesion of the TiO2 films.
[1] Kalisz, M.; Grobelny, M.; Świniarski, M.; Mazur, M.; Wojcieszak, D.; Zdrojek, M.; Judek, J.; Domaradzki, J.; Kaczmarek D. Comparison of structural, mechanical and corrosion properties of thin TiO2/graphene hybrid systems formed on Ti-Al-V alloys in biomedical applications. Surf. Coat. Tech. 2016, 290, 124–134, DOI: 10.1016/j.surfcoat.2015.08.011.
Point 2: What would be the mechanism of the corrosion behavior of Mg with a thicker TiO2 film? Where the body fluids go if a high quality uniform thick TiO2 is covered?
Response 2: The authors want to thank you for the remark. Although, the thicker TiO2 film is uniform, it is porous (porosity is equal to 2.91 %; measurements of porosity were added into the manuscript according to Reviewer 1 comments). The chloride ions in Ringer’s solution (8.6 g/dm3 NaCl, 0.3 g/dm3 KCl, 0.48 g/dm3 CaCl2 . 6H2O) penetrate the TiO2 coating and reach the substrate through the micropores. Increase in immersion time causes that the TiO2 film begins to degrade. Thus more solution interact with the Mg alloy. When TiO2 coated alloy is immersed in Ringer’s solution, the MgO in the outer layer starts to react with chloride-containing solution, and converts to Mg(OH)2. According to FTIR results, the main corrosion reactions between the magnesium alloys with the TiO2 films and the chloride solution were associated with the formation of hydroxides and carbonates. It can be assumed that Mg(OH)2 were formed first, and then the reaction between them and CO2 from the air resulted in the formation of MgCO3.
Point 3: It would be great if the authors show how the corrosion rates are changed with different thickness of films? With the hydrogen evolution reduction, there should be delayed Mg corrosion based upon the references cited in Page 10.
Response 3: We agree with the Reviewer’s opinion. The corrosion rates measured by the volume of evolved H2 changed with different thickness of the TiO2 films, and they decrease with reduction of hydrogen evolution volume. The corrosion rates, vcorr, of the films are presented in Table 1.
Table 1. Corrosion rates calculated from the hydrogen evolution volume after 24 hours of immersion for MgCa2Zn1 and MgCa2Zn1Gd3 alloys, and the TiO2 films applied.
|
Sample |
Corrosion rate, vcorr, mm·y-1 (after 24 h) |
|
MgCa2Zn1 |
4.9 ± 0.02 |
|
TiO2 (200 nm thick)/MgCa2Zn1 |
3.08 ± 0.02 |
|
TiO2 (400 nm thick)/MgCa2Zn1 |
4.34 ± 0.02 |
|
MgCa2Zn1Gd3 |
6.45 ± 0.02 |
|
TiO2 (200 nm thick)/MgCa2Zn1Gd3 |
4.81 ± 0.02 |
|
TiO2 (400 nm thick)/MgCa2Zn1Gd3 |
2.4 ± 0.02 |
Point 4: The cell culture study shows the improved cell viability with a thicker TiO2 film, which is well known outcomes from many previous studies. It would be great how the cell morphologies look like and how this study has its own originality.
Response 4: The authors want to thank you for this remark. The corrosion is the major problem in compatibility of orthopedic implants. The evolution of hydrogen is a key issue to ensure the TiO2 coated alloy is biocompatible for the use in biomedical implants. So, the authors wanted to perform the initial test in biocompatibility which is cytotoxicity.
The results showed that density of cells cultured on the TiO2 film 400 nm thick applied onto MgCa2Zn1GD3 alloy is compared to control. The cells had proper, characteristic to fibroblasts, spindle-shaped morphology.
A slightly lower cell density was observed for the MgCa2Zn1Gd3 alloy, the morphology of cells seeded on this substrate was also correct.
In the case of material with MgCa2Zn1 with 200 nm thick TiO2 film, changes in cells morphology and proliferation were shown. The cell density was low, most of the cells were rounded, what indicates their degradation.
The above observations are consistent with cell viability data.
